# Mitochondria Content and Activity Are Crucial Parameters for Bull Sperm Quality Evaluation

**DOI:** 10.3390/antiox10081204

**Published:** 2021-07-27

**Authors:** Zofia E. Madeja, Marta Podralska, Agnieszka Nadel, Marcin Pszczola, Piotr Pawlak, Natalia Rozwadowska

**Affiliations:** 1Department of Genetics and Animal Breeding, Faculty of Veterinary Medicine and Animal Sciences, Poznan University of Life Sciences, Wolynska 33, 60-637 Poznan, Poland; marcin.pszczola@up.poznan.pl (M.P.); piotr.pawlak@up.poznan.pl (P.P.); 2Institute of Human Genetics, Polish Academy of Sciences, Strzeszynska 32, 60-479 Poznan, Poland; marta.podralska@igcz.poznan.pl (M.P.); agnieszka.nadel@igcz.poznan.pl (A.N.); natalia.rozwadowska@igcz.poznan.pl (N.R.)

**Keywords:** spermatozoa, sperm motility, computer-assisted sperm analysis (CASA), mitochondria content and membrane potential, gene expression, oxygen consumption rate (OCR), Seahorse XF ATP Agilent

## Abstract

Standard sperm evaluation parameters do not enable predicting their ability to survive cryopreservation. Mitochondria are highly prone to suffer injuries during freezing, and any abnormalities in their morphology or function are reflected by a decline of sperm quality. Our work focused on describing a link between the number and the activity of mitochondria, with an aim to validate its applicability as a biomarker of bovine sperm quality. Cryopreserved sperm collected from bulls with high (group 1) and low (group 2) semen quality was separated by *swim up*. The spermatozoa of group 1 overall retained more mitochondria (*MitoTrackerGreen*) and mtDNA copies, irrespective of the fraction. Regardless of the initial ejaculate quality, the motile sperm contained significantly more mitochondria and mtDNA copies. The same trend was observed for mitochondrial membrane potential (ΔΨm, JC-1), where motile sperm displayed high ΔΨm. These results stay in agreement with transcript-level evaluation (real-time polymerase chain reaction, PCR) of antioxidant enzymes (*PRDX1*, *SOD1*, *GSS*), which protect cells from the reactive oxygen species. An overall higher level of glutathione synthetase (*GSS*) mRNA was noted in group 1 bulls, suggesting higher ability to counteract free radicals. No differences were noted between basal oxygen consumption rate (OCR) (Seahorse XF Agilent) and ATP-linked respiration for group 1 and 2 bulls. In conclusion, mitochondrial content and activity may be used as reliable markers for bovine sperm quality evaluation.

## 1. Introduction

Gamete quality (sperm and oocytes) is one of the critical factors directly determining reproductive success both in vivo and in vitro. The role of sperm in animal reproduction is not solely restricted to providing the haploid male genome to the oocyte. At the time of fertilisation, it delivers several components, including a versatile population of RNAs. Standard semen analysis (applied both for human and animal diagnostics) rely on sperm concentration count, morphology, and motility assessment. Yet, these parameters are not universal for predicting fertilisation success. It was shown that about 50% of couples undergoing an IVF (in vitro fertilisation) programme with failed fertilisation had normal pre-IVF semen analysis [1,2]. A similar case has been reported for the artificial insemination (AI) bulls, for which the computer-assisted sperm analysis (CASA) results exceeded the minimal accepted values and showed over 92% motile sperm of normal morphology and DNA integrity. Frozen/thawed sperm had good total and progressive motility; however, AI of 16 heifers resulted in no pregnancies, and the blastocyst production rate was nearly identical to that produced using the dead sperm [3]. With a growing understanding of sperm function and development of new diagnostic approaches, in an increasing number of cases, the underlying causes of “*idiopathic*” infertility may be now diagnosed. The main causes will oscillate within the genetic/epigenetic factors, sperm-carried RNA, sperm proteome, sperm ultrastructure, centrioles, and mitochondria (mt).

Recent research in animal and human sperm physiology is placing an increasing interest on the mitochondrion as a biomarker of sperm health and fertility [4,5,6,7]. As the energy “*power plants*” of a cell, the role of mitochondria for male fertility is directly linked to motility, but these organelles are also crucial for sperm hyperactivation, capacitation, acrosin activity, acrosome reaction, and DNA integrity. Mitochondria produce physiological levels of reactive oxygen species (ROS), which address redox-related cell signalling pathways [8,9,10]. Elevated ROS production contributes to mitochondrial damage in a range of pathologies [11]. Mitochondrial DNA (mtDNA) is also susceptible to oxidative damage, and mutations in mtDNA may compromise sperm function [12,13]. Moreover, mitochondrial dysfunction has been implicated in the pathogenesis of seminal oxidative stress (OS), which is a key element responsible for many cases identified as *idiopathic* male infertility [14]. Studies showed that human sperm subjected to OS in the male reproductive tract had reduced motility and fertilising potential [15]. Exposing bull semen to epicatechin, a natural flavonoid that also exhibits powerful ROS scavenging and metal-chelating properties, allowed reversing ROS-mediated alterations to motility, viability, DNA integrity, and oxidative profile in cryopreserved spermatozoa [16]. Impaired mitochondrial function severely jeopardises the maintenance of energy production required for sperm motility and may be an underlying cause of asthenozoospermia [17]. A high percentage of patients with severe asthenozoospermia showed increased basal and stimulated ROS production, increased mtDNA copy number, decreased DNA integrity, and low mitochondrial membrane potential (ΔΨm). A smaller cohort of patients also showed nuclear DNA fragmentation [17].

Given that the potential for cell movement is dependent on metabolic energy resulting from mitochondrial functions, evaluation of mitochondrial membrane potential may be indispensable to predict the quality of sperm motility [18]. ΔΨm is generated by the proton pumps and is an essential component in the process of energy storage during oxidative phosphorylation (OXPHOS). Together with the proton gradient (ΔpH), ΔΨm forms the transmembrane potential of hydrogen ions, which is harnessed to make ATP. Therefore, it may serve as a sensitive indicator for the energetic state of the mitochondria and the cell, and it can be used to assess the activity of the mitochondrial respiratory chain, electron transport chain, and the activation of the mitochondrial permeability transition [19]. Another important measure of mitochondrial function is provided by the oxygen consumption rate (OCR). It reflects the electron transport rate and stands as equivalent of OXPHOS during which oxygen is consumed by the cell. Bovine sperm mostly obtain energy vis OXPHOS [20]. Thus, measuring changes in concentrations of dissolved oxygen and free protons enabled by the *Agilent Seahorse XF* analysers may be applied to evaluate sperm mt-respiration.

Fundamentally, the sperm mitochondrial structure and function are similar to that of the somatic cells; however, these organelles do show some morphological and functional changes throughout spermatogenesis. Flow cytometry analysis revealed variations in DNA content, chromatin structure, and dramatic changes in the mitochondria of mouse germ cells during maturation from spermatogonia to elongated spermatids [21]. Changes in mitochondrial protein composition have been noted during bull and mouse spermatogenesis [22]. Spermatogonia and early primary spermatocytes harbour “*somatic cell type*” mitochondria; late primary and secondary spermatocytes and early spermatids contain “*condensed*” and metabolically more efficient mitochondria. Late spermatids and mature spermatozoa have an intermediate-type organelles [23]. Studies revealed that spermatocytes require mitochondrial fusion as they undergo an acute upregulation of OXPHOS during meiotic prophase I, thus demonstrating how this process safeguards mitochondrial function and male fertility [24].

During spermatid differentiation, a subset of mitochondria line the sperm midpiece, while the rest undergo phagocytic degradation in the Sertoli cells [25]. The mechanism behind this process still remains unknown [23]. According to the literature, anything between 50 and 75 mitochondria remain anchored around the nine outer dense fibres of the axoneme. During late spermiogenesis, spherical mitochondria gather around the forming spermatid tail, finally become organised end-to-end, and wrap helically around the flagellum, forming the thick mitochondrial sheath [26]. This structure is stabilised by disulphide bonds between the cysteine and proline-rich proteins; it further protects mitochondria (and mtDNA) and provides resistance to the hypo-osmotic stress. Scanning and transmission electron microscopy (SEM and TEM) studies of sperm from 10 mammalian species (human, mouse, rat, dog, rabbit, goat, pig, bull, guinea pig, and golden hamster) revealed the relationship between sperm internal structures, flagellar length, sperm physiology, and also indicated that mitochondrial volumes were positively correlated with ATP content and sperm swimming velocities [27]. TEM examination allowed describing a midpiece structural defect in Charolais AI bull, where 24–36% of frozen/thawed sperm cells manifested mitochondrial aplasia at the neck region, mitochondrial segmental elongation and gaps and thickening of the outer dense fibers at the apical region of the midpiece, and loss of the cementing substance and development of plasma membrane extensions in the entire midpiece [28].

Concluding from the available data, it is evident that standard methods of semen evaluation relying on concentration, movement, morphology, and sperm cell membrane integrity analyses do not allow drawing full conclusions on the ability of a sperm cell to fertilise an oocyte. Except for human-assisted reproduction, cattle is the only animal species for which insemination with cryopreserved semen, IVF, and embryo production are done on a large scale [29]. Therefore, data obtained from studies on bull semen may also provide useful information to the clinicians. Advances in semen/sperm evaluation methods directly correlate to the success of commercially applied bio-techniques such as semen freezing (cryopreservation), insemination, and embryo production. The level of energy requirement of a cell is reflected in its ability to survive cryopreservation [30]. Out of all cell’s organelles, mitochondria are the most prone to suffer injuries during freezing. Any morphological or functional abnormalities of mitochondria are directly linked with a decrease in sperm motility and its ability to fertilise. Despite intensive studies of mitochondrial function and structure, not many works relate to functional correlations, and the available data are incoherent. In addition, not much published data exist that investigate how the mitochondrial content and activity correlate with the classical markers of sperm quality. Thus, the aim of this work was to search for a link between the number and the activity of mitochondria located in bull spermatozoa, with the goal of creating a new tool allowing to distinguish between good and poor quality spermatozoa.

## 2. Materials and Methods

### 2.1. Material Collection and Preparation for Analyses

Bull semen was obtained from the local centre for bull breeding and artificial insemination: *Wielkopolskie Centrum Hodowli i Rozrodu Zwierząt w Poznaniu*, *Poznańska 13*, *63-004 Tulce*, *Poland (WCHiRZ)*. Semen samples were prepared according to the standard AI protocol for cryopreservation, packed in AI straws, frozen, and stored in liquid nitrogen. For each male, all of the straws were obtained from the same batch (production day) of collected semen. To secure comparable initial concentrations of sperm, one straw was used per single experimental procedure; i.e., one strew was thawed for the *Mitotracker Green* procedure, one was thawed for the JC-1 staining, one was used for RNA extraction, and one was used for mtDNA analysis. The experiment was performed on 2 groups of animals: (1) bulls of high AI performance (*n* = 7) and sperm quality (evaluated by standardised semen evaluation protocol for the AI bulls), and (2) AI bulls (*n* = 10) that were temporarily disqualified from semen production, due to severely decreased parameters such as ejaculate volume, sperm concentration, and motility. These bulls were also subjected to veterinary examination, 5/10 animals revealed inflammation-related problems in testes or accessory glands. The remaining 5 animals did not have medical symptoms but had long intervals between semen collection and less physical activity. In all cases, the semen samples were additionally subjected to the standardised CASA analysis, which confirmed the results of standard semen analysis performed at the AI station. The CASA analysis was performed with Sperm Class Analyser^®^ System (Microptic, Barcelona, Spain) with a module set for frozen/thawed bull semen.

Prior to analysis, the AI straws were thawed (water bath, 37 °C), and the spermatozoa were cleared off the cryoprotectants by washing in 500 μL of BoviWash^TM^ (BW-100, Nidacon, Mölndal, Sweden) followed by 3 min centrifugation at 300× *g*. The supernatant was removed, and the remaining pellet (sperm) was used for further procedures.

*Swim up* was done according to the standardised protocol [31] with some modifications. After the initial wash, the sperm pellet was re-suspended and washed 1× in 300 μL of Sperm-TALP medium, which was followed by a brief centrifugation (3 min, 500× *g*). The obtained pellet was layered under 300 μL Sperm-TALP medium (containing Heparin, 1 μg/mL) in an Eppendorf vial, positioned at an angle of 45°, and placed for 60 min in an incubator (38 °C, 5% CO_2_). Finally, the upper fraction (containing motile, live sperm) was gently separated from the lower fraction.

### 2.2. Flow Cytometry (Mitochondria Content and Activity)

#### 2.2.1. Mitochondria Content per Sperm Cell (MitoTracker^TM^ Green FM)

The staining was done according to the manufacturer’s protocol (M7514, Thermo Fisher Scientific, Waltham, MA, USA). Prior to the procedure, the sperm were fixed in 2% PFA in PBS (15 min, 37 °C), washed in phosphate buffered saline (PBS), and centrifuged (1 min, 1200× *g*). Then, the pellet was suspended in 200 μL *MitoTracker Green* dye solution (20 nM in Hepes) and incubated for 60 min at 37 °C.

#### 2.2.2. Mitochondria Activity (JC-1)

The staining was done according the manufacturer’s protocol (T3168, Invitrogen/Thermo Fisher Scientific, Waltham, MA, USA). The sperm samples were incubated for 60 min at 37 °C in 300 μL Sperm-TALP medium containing 8 μM of the JC-1 dye. After incubation, the samples were briefly washed in PBS to remove the excess of the dye (1 min, 1200× *g*).

The specificity of fluorescent staining of the mitochondria was validated by confocal imaging by a laser scanning confocal microscope Zeiss LSM 880 and super-resolution (SR) Airyscan system Zeiss (Jena, Germany). Filter sets used included a 488 nm filter with band pass 491–553 nm (laser Argon2), 410–480 nm DAPI (laser Diode 405), and 543 nm with band pass 584–632 nm (laser HeNe1), and the Plan-Apochromat 63×/1.4 Oli DIC M27 objective was used.

#### 2.2.3. Flow Cytometry

The analyses were performed on Amnis FlowSight (Luminex, Austin, TX, USA), separately for the *MitoTracker Green* and the JC-1 dyes. *MitoTracker Green* dye (490/516 nm excitation/emission) was excited by the 488 nm laser, and the fluorescence signal was detected in channel 2 (532/555 nm). In case of JC-1 (which is a fluorescent lipophilic carbocyanine dye used to measure mitochondrial membrane potential), the fluorochrome was also excited by the 488 nm laser. At high ΔΨm, JC-1 forms complexes known as J-aggregates, which emit an orange–red fluorescence with a maximum at 595 nm (emission signal detected in channel 4, 610/630 nm), at low ΔΨm, JC-1 remains in the monomeric form emitting green fluorescence with a maximum at 530 nm (signal detected in channel 2, 532/555 nm). The final ΔΨm of a cell is established as a ratio between the red and the green fluorescence.

Flow cytometry acquisition speed was set up to low speed and the highest resolution. The acquisition was performed at 20× magnification using the INSPIRE software. Approximately 30,000 single cell events were acquired from each bull sperm sample suspended in 100 μL PBS. The obtained data were analysed using the IDEAS software. First, the cells (spermatozoa) were gated from debris using a plot of bright-field channel 1 “Area” versus channel 1 “Aspect Ratio” (Figure 1). Next, the histograms showing signal intensities were generated for the particular dyes. Finally, the data were exported for statistical analysis.

### 2.3. Quantitative Gene Expression Analysis (Q-PCR)

#### 2.3.1. RNA Extraction and cDNA Synthesis

RNA was extracted with the RNeasy Micro Kit (74004, Qiagen, Hilden, Germany) according to the manufacturer’s protocol, with important modifications at the cell lysis step. Efficient lysis is crucial for successful RNA isolation from spermatozoa. Sperm cells (especially bovine) are particularly resistant to the standard lysis conditions of the commercially available kits (cells do not lyse, even after extended incubation and vortexing). Moreover, in case of semen, there is always a danger that the sample might contain a certain amount of the somatic cells. After thawing and initial washing, the sperm samples were incubated in 500 μL of the Somatic Cell Lysis Buffer (SCLB: 0.05% SDS, 0.25% Triton X-100 in DEPC H_2_O) for 10 min on ice, followed by 1× wash in PBS (500 μL) and centrifugation at 1500× *g*, buffer based on Bianchi, et al. [32]. Next, 350 μL of the RLT lysis buffer (from the Qiagen kit, Hilden, Germany) was added, the samples were vortexed for ≈5 min, and 10 μL of Proteinase K (stock 20 mg/mL) was added (10 min incubation, 55 °C). During the incubation, the samples were briefly vortexed (5× times) and finally transferred on to the microbeads (BeadTubeDry 0.5 mm, 180620, EURx) for homogenisation (5 min, top speed, Qiagen TissueLyser LT). From this point, the homogenised lysates followed the kit’s extraction protocol. The concentration of the extracted RNA was measured by NanoDrop c2000 (Thermo Fisher Scientific, Waltham, MA, USA), and the values ranged between 31.5 and 45.9 ng/mL for samples obtained from 1 insemination straw (≈1.5 × 10^6^ sperm cells) and 8 to 21 ng/mL for fractions separated by the *swim up* procedure. Additionally (at the optimisation step), to verify the efficiency of our extraction protocol, RNA integrity (RIN) was measured with Agilent 2100 Bioanalyzer using Agilent RNA 6000 Pico chips. The obtained the RIN values were close to 3, which is indicative and characteristic for high-quality sperm [33,34]. The extracted RNA was stored at −80 °C.

Reverse transcription was done from total RNA with a Transcriptor High Fidelity cDNA Synthesis kit (5091284001, Roche, Basel, Switzerland) according to the manufacturer’s protocol. cDNA was synthesised using both Random Hexameres and OligodT. Samples were stored at −20 °C.

#### 2.3.2. Quantitative PCR (Q-PCR)

The abundance of transcripts coding the important antioxidant enzymes: superoxide dismutase (*SOD1*), peroxiredoxin (*PRDX1*), and glutathione synthetase (*GSS*) was evaluated for each of the experimental groups. The reactions were carried out on a Roche Light Cycler 96 instrument (Roche, Basel, Switzerland). Glyceraldehyde 3-phosphate dehydrogenase (*GAPDH)* and β-actin (*ACTB)* served as the reference genes. TaqMan probes and primers (sequences listed in Table 1) were synthesised by TIB MOLBIOL (Berlin, Germany), and the Q-PCR reactions were done with the FastStart Essential DNA Probes Master (06402682001, Roche, Basel, Switzerland) according to the manufacturer’s protocol. The reaction mix contained primers (0.2 µM), probe (0.3 µM), and 1 µL cDNA. The reaction conditions included pre-incubation (95 °C/600 s) followed by 45 cycles of amplification (95 °C/15 s) and annealing (59 °C/40 s). Quantification was done based on standard curves generated for each gene as a series of 10-fold dilutions of known concentration. Each sample was analysed in two independent replicates, and the mean value was used for calculations. The relative mRNA content was calculated to the normalised mean transcript level of the reference genes.

### 2.4. mtDNA Evaluation

DNA was isolated with a High Pure PCR Template Preparation Kit (11796828001, Roche, Basel, Switzerland) according to the manufacturer’s protocol. Sperm samples were lysed with Tissue Lysis Buffer (70 °C, 10 min) and incubated with Proteinase K and Binding Buffer (55 °C, 10 min). The following steps included serial washes on filter columns and elution of total DNA (with 150 μL of the Elution Buffer), according to the protocol provided. All centrifugation steps were done at 7500× *g* for 30 s. Samples were eluted to the DNA LoBind tubes (Eppendorf, Hamburg, Germany) and frozen at −20 °C. DNA concentration was measured with NanoDrop c2000 (Thermo Fisher Scientific, Waltham, MA, USA).

A real-time PCR and absolute quantification module was used to calculate the number of copies of mtDNA per single spermatozoa. Quantification was done based on the standard curve method. A series of 10-fold dilutions of known concentrations (converted into the “number of molecules”, as previously described [35]) were prepared for the mitochondrial cytochrome b gene (*CYTB*) gene and nuclear β-actin gene (*ACTB*). Primer and TaqMan probe sequences (TIB MOLBIOL, Berlin, Germany) are listed in Table 1. Primer and probe concentrations were 0.5 and 0.2 μM, respectively, and 3 μL of DNA was added to each PCR reaction. Q-PCR reactions were performed on Roche Light Cycler 480 instrument using a Light Cycler 480 Probes Master kit (04707494001, Roche, Basel, Switzerland). The reaction steps included initial denaturation (95 °C, 300 s), 40 amplification cycles of denaturation (95 °C, 10 s), annealing (60 °C, 30 s), and extension (72 °C, 1 s). All samples were analysed in technical duplicates. We used Roche software for the absolute quantification of the *ACTB* and *CYTB* gene copy numbers. The *CYTB* copy number per single sperm cell (which is considered as being equivalent of the mtDNA copy number) was established as a ratio of *CYTB* expression level to *ACTB* gene expression.

### 2.5. Oxygen Consumption Rate Analysis (OCR)

#### 2.5.1. Agilent XF Seahorse Analysis (Mitochondrial Respiration Analysis)

For oxygen consumption rate and ATP-linked respiration evaluation, the *Seahorse XF Cell Mito Stress Test* was applied. The compounds used during the measurements enabled the assessment of the key parameters of mitochondrial function. As there in no commonly used protocol for OCR/ATP-linked respiration and overall mitochondrial function evaluation in spermatozoa, the extensive optimisation of the procedure was performed (data not shown). Among the optimised elements, cell concentration from 10^5^ to 10^6^ per well, a well coating matrix (fibronectin versus ConA), sperm capacitation (Sperm-TALP +1 μg/mL Heparin pre-incubation), energy source (glucose vs. fructose), and a broad range of carbonyl cyanide 4-trifluoromethoxy-phenylhydrazine (FCCP) concentrations (0.125–2.0 μM) were tested. The below optimised protocol was chosen as the one showing the minimal divergence, and the sperm cell response was the most reliable.

#### 2.5.2. Preparation of Bovine Sperm Cells

Semen samples (group 1 and 2) recovered from the AI straws were thawed in a 37 °C water bath for 1 min and diluted in 1 mL of Sperm-TALP medium. The sample was centrifuged at 150× *g* for 5 min. The pellet was re-suspended in Agilent Seahorse XF Base Medium (102353-100, Agilent, Santa Clara, CA, USA) with 1 mM pyruvate, 2 mM glutamine, and 10 mM glucose, pH was adjusted to 7.4.

#### 2.5.3. Seahorse XF Cell Mito Stress Test Kit

First, 8 × 10^6^ sperm cells in 50 μL were seeded per well of 8-well XF Seahorse Cell Culture Microplates (Seahorse Bioscience, Santa Clara, CA, USA) coated with fibronectin (1 mg/mL, Sigma F1141). To ensure spermatozoa attachment, 70× *g* plates were centrifuged for 1 min, and subsequently, 150 μL of assay medium (Agilent Seahorse XF Base Medium, 102353-100, Agilent, Santa Clara, CA, USA) per well was added. The prepared microplates were incubated for 20 min at 37 °C.

The metabolic analysis was performed according to the procedures described in the *Seahorse XF Cell Mito Stress Test* kit (Agilent, Santa Clara, CA, USA). After the sensor cartridge calibration, the microplate was placed in the Seahorse XF device, and the basal respiration was detected as baseline OCR. During the next step of the analysis, the OCR was measured after sequentially automatic injections of (1) Oligomycin A (1.5 µM final concentration), (2) FCCP (1 µM final concentration), and (3) Rotenone (0.5 µM final concentration).

For proof-of-concept analysis (control of the assay), the A549 human eukaryotic cell line was analysed. The scheme of A549 analysis mirrored the protocol applied for sperm cells with one exception; 4 × 10^4^ cells were seeded per well the day before the analysis to ensure proper cell adhesion to the plastic cell-treated surface of the microplate.

### 2.6. Statistical Analysis

#### 2.6.1. CASA Analysis

Due to the lack of normal distribution, the significance of differences between group 1 and group 2 bulls was assessed using two-sample Wilcoxon test available in rstatix package [36]. The *p*-values were adjusted for multiple comparisons using Benjamini–Hochberg correction method.

#### 2.6.2. Flow Cytometry

The results obtained by flow cytometry (*MitoTracker Green* and *JC-1 staining*), for the studied groups of bulls of high (1) and low (2) sperm quality parameters (alongside the corresponding *swim up* fractions) were subjected to statistical analyses. Each time, the values of fluorescence intensity detected for single spermatozoa (per experimental group/bull/*swim up* fraction) were used for calculations. Differences in fluorescence intensity between these groups were assessed using mixed models equations with the random bull effect, as analysing multiple sperm cells per bull resulted in repeated observations for each individual. The statistical analyses were performed in the lme4 package [37] available in R environment [38]. The significance of the differences between the groups was assessed using estimated marginal means implemented in Emmeans package [39]. Separate analyses were performed for *MitoTracker Green* and JC-1 stainings. Prior to the analyses, the fluorescence intensities were normalised to meet the assumptions of the applied statistical analyses. Fluorescence intensities obtained for the *MitoTracker Green* staining were transformed by taking the square root of the original values. In case of JC-1, the ratio of red (emission detection 610/630 nm) to green (emission detection 532/555 nm) fluorescence, which represents the state of the mitochondrial condition (ΔΨm) was log-transformed and used in further analyses. For analyses, the following general model was used:(1)$$y_{ijkl}=status_{i}+swimup_{j}+bull_{k}+e_{ijkl},$$
where y is the square root of intensity detected per single sperm cell after the *MitoTracker Green* staining, or a natural logarithm of a ratio between the red and the green fluorescence detection channels for the JC-1 dye, recorded for the *l* single spermatozoa. The status
***i*** was a fixed effect of the ***k****-th* bull’s sperm quality parameters that could take two values (high-quality (group 1) or poor-quality (group 2)) and swimup
***j*** was a fixed effect indicating whether the evaluated sperm cell originated from the upper or the lower *swim up* fraction. The bull represents the random effect of bull ***k***, and e was the random error term. All data with *p* ≤ 0.05 were considered statistically significant.

#### 2.6.3. Q-PCR and mtDNA Evaluation

The analyses were performed using the IBM SPSS Statistics 25.0. All data (before computing) were subjected to testing for normal distribution using the Kolmogorov–Smirnov and Shapiro–Wilk tests. The differences in the mRNA levels were analysed using the nonparametric two-tailed Mann–Whitney U test. The differences in the mtDNA copy number between the experimental groups were tested using the Kruskal–Wallis and the two-tailed Mann–Whitney U tests. All data with *p* ≤ 0.05 were considered statistically significant.

## 3. Results

### 3.1. Sperm Motility and Morphology (CASA Analysis)

As described in the M&M section, the semen samples were collected from two groups pf bulls: (1) high AI performance and sperm quality and (2) temporarily disqualified from semen production, due to severely decreased semen/sperm parameters. The initial choice of animals was based on the standard semen evaluation protocol and the veterinary examination done at the WCHiRZ centre (data not shown). Since the initial laboratory evaluation was done on fresh semen, and the assessed parameters included only ejaculate volume, sperm concentration, movement assessment, and morphology, CASA analysis on frozen/thawed sperm was done to validate the observed differences between the experimental groups (Table 2). CASA evaluation proved the results obtained in the laboratory. Group (1) fulfilled the regulations of the Ministry of Agriculture (Poland) for bull post-thaw semen. Over 52% of the analysed sperm cells were motile (norm is 50%) and 38.8% displayed progressive movement (PM). The cut-off value for PM of CASA analysed frozen/thawed bull semen was suggested to be 15% [40]. Most (89.3%) of the analysed spermatozoa had proper morphology (norm is 80%), 0.1–5.3% had morphological abnormalities such as coiled tail (0.1%), bent tail (1.3%), Dag defect (1.7%), distal and proximal cytoplasmic droplets (2.5 and 5.3% respectively). Semen collected from group (2) had only 30.2% motile sperm, 14.9% had proper progressive movement, 78.05% had normal overall morphology, Dag defect was diagnosed in 4.8% of the analysed cells, and 3.85% of spermatozoa had bent tails, 0.2% had coiled tails, and distal and proximal droplets were observed in 7.0 and 6.8% of the analysed cells. These values represent the median for each of the two analysed groups of bulls. Detailed mean values of sperm motility parameters (DMR, ALH, BCF, DAP, DCL, DSL, LIN, VAP, VCL, VSL, WOB, and STR), together with abbreviation explanations and statistical significance may be found in Table 2. Except for ALH and BCF, all values differed significantly within a range of *p* ≤ 0.05 and *p* ≤ 0.008.

### 3.2. Total Mitochondrial Content (MitoTracker Green Staining and mtDNA Copy Number)

It may be speculated that a certain (optimal) mitochondria content (mt number per individual spermatozoa) is needed for a sperm cell to secure its ability to fertilise the oocyte. For this reason, we have applied mt specific fluorescent staining (*MitoTracker Green*), which allowed evaluating the overall mt number per single cell, based on the level of emitted fluorescence. The specificity of fluorescent labelling was validated by confocal imaging (Figure 2).

The results indicated that on the average, spermatozoa of bulls with high sperm evaluation parameters (group 1) contained 3.102 times more mitochondria (fold change, *p* ≤ 0.01) than sperm cells of bulls assigned to group 2 (Figure 3A). To verify whether the accumulation of mitochondria within the midpiece of spermatozoa corresponds to its motility, semen were subjected to *swim up*. The results indicate that regardless of the studied bull group (high semen parameters versus low), the motile (upper) fraction contained significantly more mitochondria (*p* ≤ 0.01) than the lower, non-motile fraction (Figure 3B). However, bulls with overall higher semen quality parameters (group 1) had retained significantly more mitochondria (*p* ≤ 0.01) than bulls disqualified from AI semen collection (group 2), irrespective of the fraction (Figure 3B).

The above results stay in agreement with the mtDNA copy number evaluation. On average, the upper fraction of group (1) bulls contained 27.75 copies of mtDNA, and 17.90 copies were calculated for the lower fraction. Similarly, in group (2) bulls, 21.01 and 10.54 copies of mtDNA were estimated for the upper and lower fraction, respectively. Generally, both the upper and the lower *swim up* fractions obtained from separating the sperm of high AI performance bulls (group 1) displayed higher mtDNA copy number (*p* ≤ 0.05) than the lower fraction of group 2 bulls.

### 3.3. Mitochondrial Membrane Potential (ΔΨm)

Similarly to the *MitoTracker Green* staining, JC-1 fluorescence detection revealed significant differences (*p* ≤ 0.01) between all of the analysed groups (Figure 3C,D). Mitochondrial membrane activity (ΔΨm) calculated as a ratio of orange–red to green fluorescence was generally higher in the spermatozoa of bulls with high semen parameters (group 1; *p* ≤ 0.01; Figure 3C). Moreover, in both groups, ΔΨm was higher in the upper *swim up* fractions (*p* ≤ 0.01, Figure 3D). The specificity of the JC-1 staining was confirmed by confocal microscopy (Figure 4).

### 3.4. Gene Expression

The level of oxidative stress is an important factor in determining male fertility. Due to the high rate of cell division and mitochondrial oxygen consumption, testicular tissue is particularly susceptible to the oxidative stress. It is critical to obtain a balance between the production of free radicals and its metabolism. Antioxidants prevent damage either by counteracting the free radicals or by preventing their formation in the testicular cells. The preventive antioxidant system incorporates enzymes such as SOD, GSS, and PRDX; thus, we have analysed and compared the level of the corresponding transcripts (Figure 5). The mRNA content of the *PRDX1* gene was not significantly different between the experimental groups; however, a 2-fold difference was observed in favour of the upper *swim up* fraction of group 1. We have detected overall less transcripts for the *GSS* gene in spermatozoa of group 2 bulls. The upper *swim up* fraction collected from bulls with high sperm evaluation parameters contained significantly more *GSS* transcripts than group 2 upper and lower fractions (*p* = 0.000 and *p* = 0.001 respectively). Moreover, even the less motile, lower *swim up* fraction collected from group 1 bulls had significantly higher *GSS* mRNA content than the spermatozoa from the upper (presumptively better quality, motile) fraction of group 2 bulls (*p* = 0.008). Significant differences were also noted when comparing the lower fractions of the two experimental groups (*p* = 0.045). An interesting observation was made of the *SOD1* transcript abundance. No statistically significant differences were noted between the two *swim up* fractions obtained for high-performance bulls (group 1), but the spermatozoa of the upper (motile) fraction of group 2 contained more *SOD1* transcript copies than the lower fraction (*p* = 0.035). Additionally, the higher *SOD1* transcript level noted for group 2 (upper fraction) was also significantly higher than the *SOD1* mRNA content in both of the fractions separated for group 1 (upper *p* = 0.004; lower *p* = 0.001).

### 3.5. Spermatozoa OCR Analysis

It was suggested that bovine spermatozoa preferentially exploit OXPHOS as the main source of energy [20]. The oxygen consumption rate may serve as a direct measure of mitochondrial electron transport rate and therefore stand as an equivalent of OXPHOS. Extracellular flux analysers (such as *Seahorse* analyser) offer a real-time assessment of cell’s metabolic state and enable tracking potential impairment of the energetic pathways. We have used standard *Seahorse* protocol for *Cell Mito Stress Test* (Oligomycin, FCCP and Rotenone) and measurements of basal respiration. ATP-linked respiration and spare respiration capacity were done. Since we have aimed to pinpoint any potential differences in the OCR between high and low sperm quality bulls, the analyses focused on samples that were not subjected to the *swim up* separation. This approach was also taken due to a large number of spermatozoa required per each experimental well. Oligomycin treatment resulted in a drop of basal respiration within 20 min of administration, but the sperm cells did not show the standard response mitochondria to the FCCP injection. In standard conditions (in somatic cells), this should be reflected by a dynamic increase of the OCR due to the collapse of the mitochondrial membrane potential. Indeed, under the same conditions, this phenomenon was detected in our control assay (diploid, metabolically active A549 cell line). An example of such analysis is shown in Figure 6. Our results showed no statistical differences in basal OCR and ATP-linked respiration values between spermatozoa obtained from bulls with high and low semen parameters (data not shown).

## 4. Discussion

The mitochondrion is being recognised as an important biomarker of sperm quality, fertilisation potential and survival after cryopreservation [6,7,41]. Effective semen cryopreservation is one of the key elements contributing to the success of assisted reproductive technologies (ART) in cattle breeding and human reproductive medicine. Published studies often make comparisons between the ejaculates of different bulls/breeds or between the *swim up* fractions [31,42,43]. While investigating the topic, we have not come across works that would compare bulls with high and low semen evaluation scores. Although, at the AI stations, a selection is being made towards animals with high fresh and post-thaw semen parameters, some bulls with superior genetic values produce semen sensitive to cryopreservation. Therefore, our experimental groups included frozen–thawed sperm collected from males with high and low semen parameters. We have aimed to describe a link between the number and the overall activity of mitochondria that could serve as an indicator of sperm quality.

In comparison to fresh ejaculate, the viability of frozen–thawed spermatozoa is still on average relatively low, and it differs significantly among the breeding bulls [44,45,46]. This is also true for other species, such as stallion [47], ram [48], and human [49]. Cryopreservation of bull (and human) sperm has outpaced other species, yet there is still room for improvement, as large portions of sperm suffers substantial damage [50].

### 4.1. Mitochondrial Content Corresponds to mtDNA Copy Number in Bovine Spermatozoa

Depending on energetic demand, cells carry a different number of mitochondria, each containing a variable number of mtDNA copies [51]. The typical mammalian sperm contains approximately 50–75 mitochondria, with 1–1.4 copies of mtDNA each [52,53]. During spermatogenesis, the mitochondrial number changes, due to fusion and a drastic loss of cytoplasm at the time of spermatid formation. Depending on the species, this corresponds to a major decline in the mtDNA copy number, 8–10-fold in mice [54] and 5–6-fold in human [55]. Some studies suggest that a certain/optimal mitochondrial content within the sperm cell secures its ability to fertilise [53]. For example, in stallion, an increased mtDNA copy number was associated with a decreased total sperm motility [56]. In human, the average mtDNA copy number was higher in infertile men and in patients with abnormal semen parameters and non-progressive sperm movement [52,57]. Yet, studies by Kao et al. [58] showed an opposite trend, where a decrease in the mtDNA content was noted in patients with asthenozoospermia or with poor sperm motility parameters (7.2 and 74.1 mtDNA copies, respectively). These results support our observations, where the spermatozoa of the motile *swim up* fraction of bulls with high sperm quality parameters (group 1) contained significantly more mtDNA copies than the sperm of the lower (non-motile) fraction of group 2 bulls (27.75 versus 10.54 copies). Moreover, this trend was also conserved within both of the lower fractions, as non-motile sperm from group 1 had on average significantly more mtDNA copies (17.90) than the lower fraction of group 2 (10.54 copies). The differences between the two upper fractions were not statistically significant, but spermatozoa of the motile fraction of group 2 bulls had less copies of mtDNA than the upper fraction of group 1. Therefore, mtDNA copy number could become an indicator of defective/proper spermatogenesis, provided that species-specific, optimal mtDNA content will be established. The functional significance of limited mtDNA copy number in sperm still remains unclear. Plausibly, it could reduce the likelihood of ROS-mediated DNA damage.

The existing contradictions in the correlation of mtDNA copy number and male fertility status may arise from several factors such as species specific differences, the use of different methods to quantify the mitochondrial genome (Southern blot vs. PCR-based assays), or contamination with other cell types (immature sperm or leukocytes) that may be present in poor quality semen [52,58]. Leukocytes, predominantly polymorphonuclear leukocytes (PMN, neutrophils), are present in most human ejaculates. Studies show that about 30% of infertile males have leukocytospermia, which is defined by the WHO as containing >1 × 10^6^ white blood cells/mL in a semen sample [59]. In our case, the sample preparation steps (*swim up* and sperm washes prior to DNA extraction) eliminate this factor.

The consistency of our data is supported by the fluorometric measure of mitochondrial content by flow cytometry. *MitoTracker Green* accumulates in mitochondria regardless of their mitochondrial membrane activity. Significant differences were noted between all of the studied groups. On average, the motile fraction of group 1 sperm had the highest mitochondrial content, which was progressively reduced in its corresponding lower fraction and in both of the *swim up* fractions obtained from poor quality sperm (Figure 3). To further validate these observations, we have evaluated the mitochondrial content in spermatozoa of high and low sperm quality bulls, without *swim up* preselection. In this case, a 3.102-fold higher mt-fluorescence signal was detected in group 1 bulls. The proportion of *MitoTracker Green* stained spermatozoa and the mtDNA content highly correlates with sperm quality and motility parameters. These results provide a coherent evidence for the applicability of both of these parameters for bovine semen evaluation.

### 4.2. Mitochondrial Membrane Potential (ΔΨm) Is an Indicator of Bull Sperm Quality

Numerous studies indicate that mitochondrial activity and function during spermatogenesis affects sperm survival. Mitochondrial respiration is required for proper spermatogenesis in mice, and defects resulting from accumulated mutant mtDNAs cause male infertility [60,61]. Mitochondria have diverse functions in testicular cells, including the production of steroid hormones, regulation of cell proliferation, and cell death (reviewed by Park and Pang [61]). ROS play a double role in this process. On one side, ROS are crucial for tyrosine phosphorylation, cholesterol efflux, and sperm–oocyte interactions, but on the other, they may induce oxidative stress and damage the spermatozoa [62,63].

Alternations in mitochondrial membrane fluidity during cryopreservation have been suggested to raise the ΔΨm and induce the release of ROS, which in turn cause single/double-strand DNA breaks [50,64,65]. Changes in osmotic balance may also induce apoptosis [50], oxidative stress [66], and loss of mitochondrial function [67]. Short-term openings of the mitochondrial permeability transition pore (mPTP), leading to the release of Ca^2+^ and ROS, fulfils physiological functions. Prolonged opening causes the collapse of the ΔΨm, ATP loss, osmotic shock, and rupture of an outer mitochondrial membrane (OMM), which eventually leads to necrosis or the release of mitochondrial pro-apoptotic factors [67,68,69]. Thus, ΔΨm may serve as an indicator of mitochondrial activity and functional capabilities of spermatozoa. ΔΨm shows a direct and significant correlation with sperm count, normal morphology, motility, and viability [70]. Spermatozoa with low ΔΨm are less capable of undergoing the acrosome reaction [71,72]. Studies by Garner and Thomas [73] have indicated that bovine sperm motility positively correlates with the number of cells exhibiting high JC-1 aggregate level. JC-1 dye serves as a mitochondrial potential sensor. It indicates the level of mt-depolarisation by assessing the red/green fluorescence intensity ratio, which reflects the ΔΨm status. Polarised mitochondria are marked by orange–red signals (JC-1 aggregates), which on depolarisation diffuse green monomer fluorescence (Figure 4). In our studies, spermatozoa collected from high AI performance bulls had higher ΔΨm than the cells obtained from bulls with impaired sperm motility and morphology parameters (Figure 3B). Studies in men showed that low sperm motility may be associated with deformations of the mitochondrial sheath containing functional mitochondria. In this case, lower ΔΨm results from the abnormal morphology of the axoneme and causes low sperm motility and severe asthenozoospermia [74].

Following the above observations, the question was raised of whether the *swim up* selected motile sperm of bulls with generally low sperm evaluation parameters (group 2) could be compared to the motile fraction of high performance bulls (group 1). Studies show that *swim up* allows successfully eliminating/separating sperm with DNA fragmentation from viable spermatozoa, significantly increasing the pregnancy success rate for sub-fertile male patients [75]. This method also allowed observing a good clinical outcome in oligospermic, teratospermic and oligo-astheno-teratospermic patients [76]. In cattle IVF, *swim up* is one of the methods of choice for selecting the best quality spermatozoa. Recently, Magdanz et al. [31] showed that spermatozoa of the upper fraction of bull sperm, except for improved motility, have higher metabolic rates and longer flagella. Our study takes this investigation further and shows that although *swim up* successfully selects motile spermatozoa with high ΔΨm, it may not level off the differences in sperm quality arising from “the male factor”. In our studies, the upper fraction from group 2 had significantly lower ΔΨm value than the upper fraction of group 1. The bottom fractions also differed, with group 2 having the lowest ΔΨm of all experimental setups. This observation may be reflecting alternations in mitochondrial dynamics caused by a disease or pathological processes affecting the testis. Studies in humans [77] and rats [78] showed that diabetes, hypoxia, varicocele, and testicular torsion [79] are associated with ROS-related mitochondrial damage and mtDNA mutations. Localised infections, systemic inflammations, or inflammatory-like conditions associated with oxidative stress may exert transient or permanent effects on male fertility. A microarray analysis of human testicular gene expression in infertile patients pointed to an increase in the expression of inflammatory-response genes [80]. Therefore, the observed overall lower ΔΨm in group 2 bulls (with or without *swim up* selection) might be reflecting the physiological status of an animal. All these males were disqualified from semen production due to severely decreased parameters such as: ejaculate volume, sperm concentration, motility, and morphology. On veterinary examination, some of these bulls revealed problems related to inflammation in testes or in accessory glands. Other males did not present any negative symptoms, but in the recent months had less physical activity or had long intervals between semen collection.

Pathological processes affecting the testis may directly affect spermatogenesis and handicap mitochondrial morphology and function. Spermatogonia and early spermatocytes contain small mitochondria with low OXPHOS activity. Later in the process, higher oxygen consumption, elevated ATP production, and downregulation of glycolytic enzymes upregulates aerobic metabolism and OXPHOS. The respiratory chain complex assembly increases and cristae develop [81]. Consequently, spermatocytes, spermatids, and spermatozoa are characterised by elongated and condensed form of mitochondria, that are able to coil around the axoneme [61]. Any defects in the above processes (including mitochondrial fusion) induce heterogeneity, reduction of mtDNA content, and OXPHOS. Depletion of mitofusins (GTPases embedded in the OMM) results in spermatogenesis arrest and a reduced number (and activity) of OXPHOS subunits [24]. All these processes are embedded within the specific testicular microenvironment, including the Sertoli and the Leydig cells. The first guard the blood–testis barrier, control the number of germ cells via apoptosis, and are a major energy source for spermatogenesis (secrete lactate and produce ATP through lipid β-oxidation) [82,83]. The latter produce steroid hormones and cAMP, which accelerates cholesterol transport to the IMM and stimulates the production of ATP in the mitochondria [84].

### 4.3. Gene Expression

Defective spermatozoa produce high levels of ROS that may also become toxic for the healthy sperm present in the semen [85]. Antioxidant enzymes such as SOD, PRDXs, or GSS can prevent this damage by counteracting free radicals. Early pachytene spermatocytes are the major site of RNA synthesis, which begins to decline around the time of mid pachytene. Transcripts are stored in spermatids and translated during sperm elongation [86]. Although it is generally believed that spermatozoa are translationally silent, some protein translation was demonstrated in human, mouse, bovine, and rat sperm at the time of capacitation [87]. It is now believed that the transcript profile of mature sperm cells may be an indicator of its quality and ability to fertilise. Whole transcriptome analysis of bovine sperm indicated associations with sperm function, fertilisation potential, and embryonic development [88].

PRDX enzymes are recognised as vital factors in male fertility, and six members of the PRDX family were detected in testis and spermatozoa [89]. They are at the first line of defence against the H_2_O_2_ and other ROS. *PRDX1* transcripts were found in human spermatogonia and round spermatids, and the corresponding enzyme was located in sperm equatorial region, nucleus, and flagella [90]. The results of our analyses revealed the presence of *PRDX1* in all of the analysed samples. Although the differences were not statistically significant, the highest (close to significance) *PRDX1* level was noted in the motile fraction of group 1 bulls. A decrease in the level of PRDXs is associated with an inability to maintain low levels of ROS. It was shown that human sperm motility parameters were significantly affected by PRDXs inhibition, the consequences of which included an increase of ROS and a decrease of ΔΨm and the intracellular level of ATP. Inhibition of PRDXs negatively affected sperm capacitation and ability to undergo the acrosome reaction. It also reduced the fertilisation rate, embryonic cleavage, and development rate in the IVF system [91,92]. PRDXs interact with several proteins related to sperm function such as serine/threonine kinase 4 (STK4), mitogen-activated protein kinase 5 (MAPK5), glutathione S-transferase (GST), sulfiredoxin 1 (SRX1), thiosulfate sulfurtransferase (TST), glutaredoxin (GRXS), and SOD1 [92].

Activity of the SOD enzymes positively correlate with sperm concentration and overall motility, but a wide range of differences was noted among various mammalian species including humans, donkey, mouse, rat, rabbit, dog, and bull spermatozoa [91,93,94,95,96]. The variable sensitivity to ROS in mammalian spermatozoa may account for the observed variations in the SODs activity. It was shown in a rat model that SODs play an important role in testicular development and spermatogenesis [97]. Genetic variation and functional polymorphism in SOD1 and SOD2 may be associated with infertility and low IVF outcome [98]. We have found higher levels of the *SOD1* gene transcript in spermatozoa collected from the motile fractions of bulls with both high and low sperm quality parameters. Studies by [94] showed that the expression of SOD1 protein was higher in spermatozoa from high fertility bulls and that SOD1 could be used to diagnose bulls based on the fertility index. Interestingly, we have noted the highest (and statistically significant) *SOD1* level in the upper *swim up* fraction from bulls with poor semen quality (group 2). This may be reflecting the negative processes happening during spermatogenesis, where spermatogonia or spermatozoa need to compensate for the rising levels of ROS by increasing the transcription of the vital antioxidant agents.

GSS is the second enzyme in the glutathione (GSH) biosynthesis pathway, and similarly to PRDX1 and SOD1 is a potent antioxidant. We have noted a drastic decrease in the *GSS* transcript level in non-motile sperm of both of the analysed groups compared to the spermatozoa of the upper *swim up* fraction obtained from high AI performance bulls (group 1). This may be supported by the observations that a significant decrease in intracellular GSH concentration was noted in seminal plasma and spermatozoa with severely impaired morphology in infertile men [99].

### 4.4. Oxygen Consumption

Mitochondrial oxygen consumption is considered the central parameter of mitochondrial function. Studies on various species have shown that mitochondrial oxygen consumption is positively correlated with traditional measures of sperm function including motility and vitality [7,31,100]. Mitochondrial oxygen consumption is a unique indicator of stallion spermatozoal health and varies with cryopreservation media [7]. Spermatozoa may use different metabolic substrates to generate energy, depending on functional requirement, such as motility, capacitation, or oocyte interaction [101]. Sperm cells obtain energy via two main pathways, OXPHOS (ATP is produced via the electron transport chain) and glycolysis (ATP is generated in the cytosol by the breakdown of sugars). Murine sperm preferentially follow the glycolysis pathway [102], and bovine sperm mainly utilised the OXPHOS pathway [20]. Studies by Magdanz et al. [31] showed that the upper (*swim up* selected), motile fraction of bovine spermatozoa has higher metabolic rates (OCR level) regardless of the bull breed. The sperm samples that were not subjected to *swim up* show almost no reaction to the administration of oligomycin, suggesting a low level of OXPHOS activity (little ATP expenditure). This could be possibly explained by the fact that hyperactivation and active movement requires higher oxygen consumption rates. In this respect, our data are consistent with these results, as we see a similar basal sperm metabolic rate over time, as presented by Magdanz et al. [31]. We saw no differences between spermatozoa of high-quality and low-quality sperm (groups 1 and 2). This is also not surprising in the context of the above data, as the differences in sperm activity reflected by the OCR may become visible only after its activation. Unfortunately, the literature is very limited, and we have not found any works on bovine sperm that would include FCCP and Rotenone treatment; thus, we have no point of reference. Works on boar sperm have showed proper (similar to the somatic cells) reaction to all of the three reagents used in the *Seahorse* assay [100]. Therefore, we believe that this part still requires intensive optimisation steps.

## 5. Conclusions

The most important conclusion arising from this work is that the parameters that allow describing the activity and the number of mitochondria in each sperm cell may be used as reliable biomarkers for bovine sperm quality evaluation. For proper sperm function, it is necessary to create a balance between produced free radicals and cell metabolism. The sperm cells of proper mobility contain highly active mitochondria (with high ΔΨm), which corresponds to higher ability to produce energy. These results stay in agreement with transcript analysis of genes coding the antioxidant enzymes, which prevent the destructive action of ROS. The diagnostic methods used in this project have the potential to become applied in animal breeding centres and as a diagnostic tool in human medicine (fertility clinics).

## Figures and Tables

**Figure 1 antioxidants-10-01204-f001:**
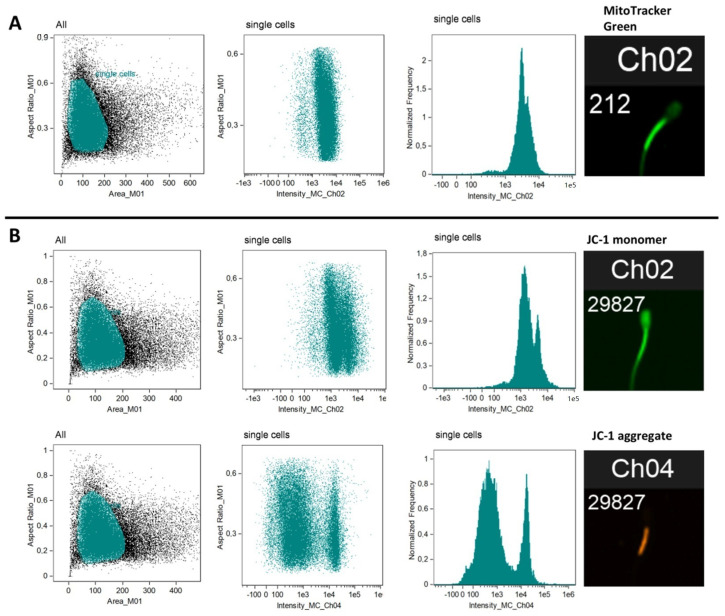
Representative images of flow cytometry data obtained for *MitoTracker Green* (**A**) and JC-1 (**B**). Images from left to right present: dot-plot graphs of the gated region (highlighted in blue) for spermatozoa—area vs. aspect ratio (signals collected in bright field); dot-plot images of signals collected at channels Ch02 (green 532/555 nm) and Ch04 (orange/red 610/630 nm); histograms representing fluorescence intensity obtained for each analysed sample (bull sperm cells); and example fluorescent images acquired for each of the analysed sperm cells.

**Figure 2 antioxidants-10-01204-f002:**
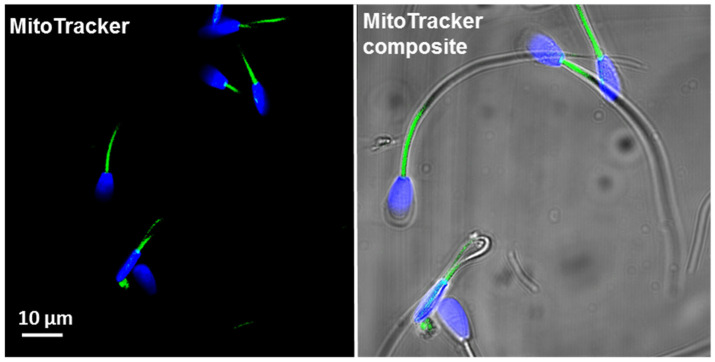
An example confocal image of spermatozoa labelled with *MitoTrackerTM Green FM*. Green signals represent the mitochondria-rich region (midpiece) and blue signals label chromatin (DAPI staining).

**Figure 3 antioxidants-10-01204-f003:**
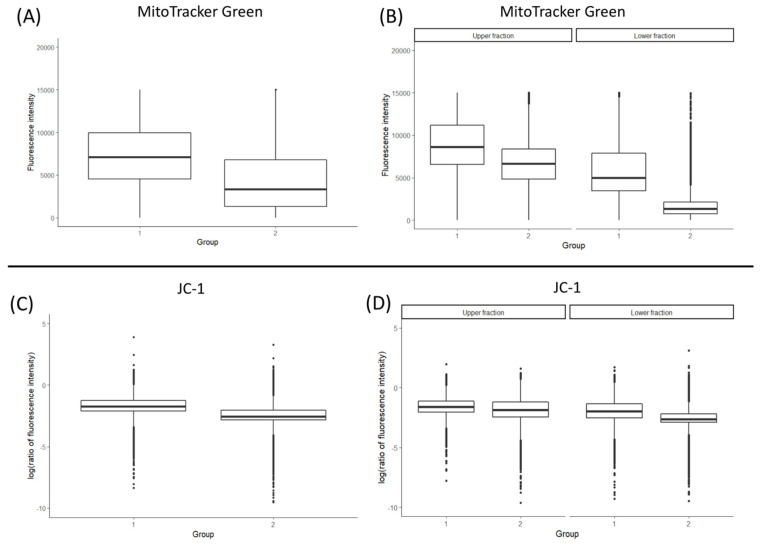
Mitochondrial content and activity in spermatozoa of bulls with high and low sperm quality parameters. (**A**,**C**) show the data obtained for the sperm samples not subjected to the *swim up* procedure, and (**B**,**D**) show the data after *swim up*. All of the observed groups differ significantly (*p* ≤ 0.01).

**Figure 4 antioxidants-10-01204-f004:**
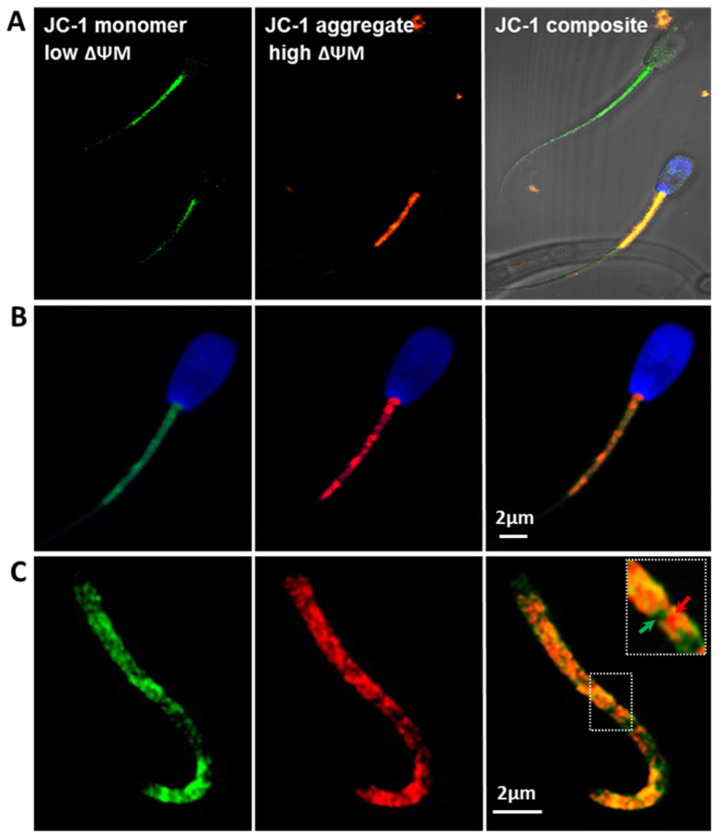
Confocal images of spermatozoa labelled with JC-1 at different magnifications. (**A**) Objective 63×; (**B**) objective 63× zoom 2×, Airyscan-SR; (**C**) objective 63× zoom 3× Airyscan-SR. Green signals label depolarised mitochondria (low ΔΨm), orange/red mark active polarised mitochondria (high ΔΨm). The images confirm that JC-1 selectively marks mt of different activity, as indicated by green (mt with low ΔΨm), and red (mt with high ΔΨm) arrows in the boxed region highlighted in panel (**C**). Blue signals label chromatin (DAPI staining).

**Figure 5 antioxidants-10-01204-f005:**
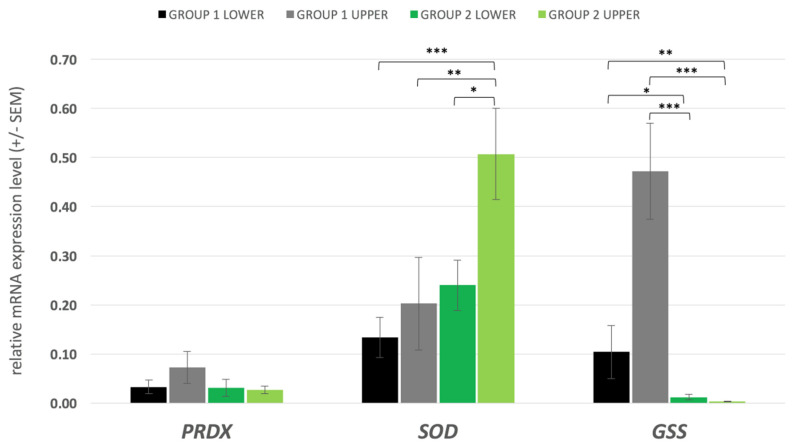
Transcript level of antioxidant enzymes genes crucial for protection against high levels of ROS in spermatozoa. “upper” and “lower” refer to the swim up fractions; statistically significant differences are indicated by *** (*p* ≤ 0.001), ** (*p* ≤ 0.01) and * (*p* ≤ 0.05).

**Figure 6 antioxidants-10-01204-f006:**
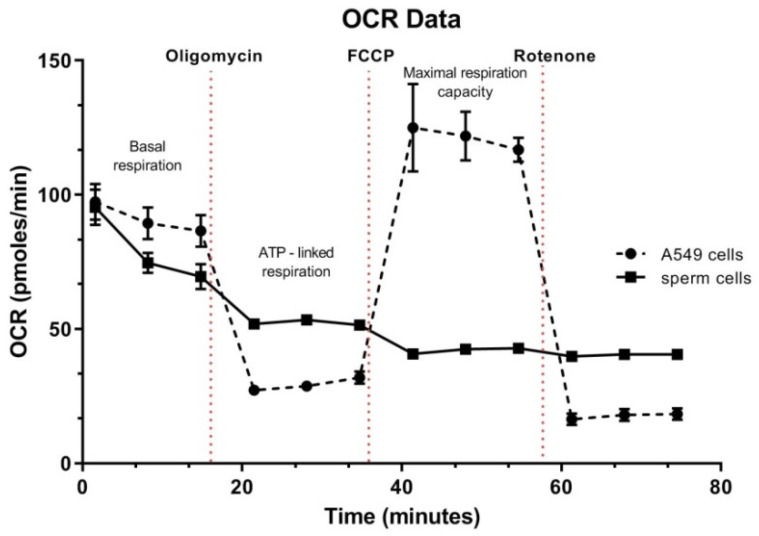
An example result of spermatozoa OCR analysis compared to the control cell line (A549). Mitochondria of the somatic cells present different reactivity to the administered chemicals (FCCP and Rotenone) than the tested sperm cells. Only oligomycin treatment results in a drop of basal respiration in sperm.

**Table 1 antioxidants-10-01204-t001:** Real-time PCR primer pairs, probe sequences, and accession number (NCBI reference sequence).

GENE	PRIMER SET	PROBE	ANNEALING (°C)	PRODUCT SIZE (bp)
**ACTB** (NCBI ref. sequence mRNA: NM_173979)	5′CCTGGGCATGGAATCCTG3′ 5′GATGTCGACGTCACACTTCATGAT3′	5′6FAM- ATTGAAGGTAGTTTCGTGAATGCCGCAG-BBQ3′	59	69
**ACTB** (DNA) (NCBI ref. sequence: NG_002441.3)	5′GGCTCGTGTGACAAAAGC3′ 5′GCAGAAGAGTGCAAGGAACA3′	5′6FAM- CAGTAGGTGCACAGTACGTTCTGAAGTGAA-BBQ3′	60	137
**CYTB** (NCBI ref. sequence: NC_006853)	5′CCCGATTCTTCGCTTTCCAT3′ 5′CTACGTCTGAGGAAATTCCTGTTG3′	5′6FAM- CATCATAGCAATTGCCATAGTCCACC-BBQ3′	60	119
**GAPDH** (NCBI ref. sequence mRNA: NC_037329.1)	5′GGCTGGGGCTCACTTGAA3′ 5′CAGGAGGCATTGCTGACAATC3′	5′YAK- TCATCTCTGCACCTTCTGCCGATGC-BBQ3′	59	134
**GSS** (NCBI ref. sequence mRNA: NM_001015630)	5′CTCCGCAGCCCTGAAAC3′ 5′ACTGAGAACATGTCGATGCACA3′	5′6FAM- AGCTGGCAGAGACGGTGTTGATTTC-BBQ3′	59	96
**PRDX1** (NCBI ref. sequence mRNA: NM_174431)	5′CCATAAACGACCTTCCTGTT3′ 5′GAGAAATATTCTTTGCTCTTCTGGAC3′	5′6FAM- CTGGAAGCCTGGCAGTGATACCATC-BBQ3′	59	159
**SOD1** (NCBI ref. sequence mRNA: NM_174156)	5′TTCCCCCGAGTCATGGC3′ 5′AGCCTTGTGTATTGTCTCCAAAC3′	5′FAM- AGTCGTGGTAACTGGATCCATTACAGGA-BBQ3′	59	177

**Table 2 antioxidants-10-01204-t002:** Median values for sperm motility parameters evaluated by computer assisted sperm analysis, CASA ^1^.

BULLS	PM (%)	Motile (%)	Bent Tail (%)	Coiled Tail(%)	Dag Defect (%)	Distal Droplet %	Proximal Droplet %	Proper Morphology %	DMR %	ALH (μm)	BCF (Hz)	DAP (μm)	DCL (μm)	DSL (μm)	LIN (%)	VAP (μm/s)	VCL (μm/s)	VSL (μm/s)	WOB (%)	STR (%)
**Group A**	38.80 ^A^ (2.588)	52.10 ^A^ (1.693)	1.30 ^A^ (0.116)	0.10 ^a^ (0.053)	1.70 ^A^ (0.130)	2.50 ^A^ (0.255)	5.30 ^a^ (0.437)	89.30 ^A^ (0.713)	1.60 ^A^ (0.127)	8.4 (0.297)	36.80 (0.776)	61.50 ^a^ (4.055)	102.3 ^a^ (6.660)	56.7 ^A^ (3.834)	58.0 ^a^ (1.817)	139.4 ^a^ (6.455)	223.6 ^A^ (13.994)	126.2 ^A^ (6.918)	64.3 ^a^ (1.355)	93.5 ^A^ (0.859)
**Group B**	14.90 ^B^ (2.227)	30.20 ^B^ (2.457)	3.85 ^B^ (0.579)	0.20 ^b^ (0.047)	4.80 ^B^ (0.628)	7.00 ^B^ (1.303)	6.80 ^b^ (1.228)	78.05 ^B^ (2.373)	4.45 ^B^ (0.399)	7.90 (0.7)	33.40 (1.061)	41.80 ^b^ (2.682)	76.50 ^b^ (8.950)	32.50 ^B^ (2.582)	47.2 ^b^ (2.142)	100.4 ^b^ (5.655)	149.25 ^B^ (8.898)	79.35 ^B^ (4.720)	54.80 ^b^ (2.259)	87.45 ^B^ (1.668)

^1^ Sperm motility parameters are: PM progressive movement, DMR distal midpiece reflex, ALH amplitude of lateral head displacement, BCF beat cross frequency, DAP distance average path, DCL distance curved line, DSL distance straight line, LIN linearity, VAP average path velocity, VCL curvilinear velocity, VSL straight line velocity, WOB wobble, STR straightness. The (%) stands for the median percentage of total. Values in brackets represent mean ± SE. Different superscripts within a column indicate significant differences (*p* ≤ 0.05; lowercase letters (a–b)) and highly significant differences (*p* ≤ 0.008; capital letters (A,B)).

## Data Availability

The data presented in this study are available in article.

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
