# Peer review of "Mitochondria Content and Activity Are Crucial Parameters for Bull Sperm Quality Evaluation"

_antioxidants, 2021, doi:10.3390/antiox10081204_

Round 1

Reviewer 1 Report

The manuscript entitled “Mitochondria content and activity are crucial parameters for bull sperm quality evaluation” is an interesting manuscript. The authors analysed the relationships between the mitochondrial activity or oxigen comsumption and the bovine sperm quality.

Introduction is a good analysis of the “state of the art”. Materials and methods are, in general, accurately described. Obtained results are clearly exposed, easy to understand and deeply discussed. 

Specific Comments

Introduction

Line 124. “Cattle is the only animal species for with AI with cryopreserved semen, IVF and Embryo production are done in a large scale”. I disagree. Nowadays horse also.

Line 129. The level of energy… (include a reference)

Line 153. Bud, change for but.

Materials and methods

Table 1. Are results (3.1). Change it. Include statistical differences between booth groups

Line 154. What CASA system did you use?

Line 156. Explain your CASA settings and analysed parameters.

Discussion

Line 491. Define ART (Assisted Reproductive Technologies)

Line 501. “Is still relatively low” in average

Line 594. Magdanz et al

Line 603. Diabetes increase the glucose levels that can increase the mitochondrial activity and ROS production. However, varicocele or testicular torsion induce a Pampiniform Plexus Blood Flow decrease resulting in a Oxygen pressure decrease. 

Line 628. Delete (rev by: Park and Pary

Line 634. And also normal spz with high motility and mitochondrial activity are producing a large amount of ROS

Line 664. What’s the reference relating SOD and sperm motility in donkey?

Author Response

The comments are included in the attached word document

Reviewer 2 Report

This manuscript addresses the study of the quality of bull semen through the quantity and activity of mitochondria.
The design of the experiment seems correct, the results are clearly stated, and the authors conduct a broad discussion. However some minor revisions should be considered before publication.
. It would be very convenient if results data were indicated in the abstract.
. In my opinion the introduction is too long and on some occasions the objective of the work is diverted to the reader. In addition, the citations referring to humans are crossed with references to bull or other species, which confuse the reader.
. Material and methods: Are several straws used per male? If so, this information would have to be provided.

Author Response

The comments are included in the attached word document.

Reviewer 3 Report

The goal of the study was to propose a method of selecting bull sperm after freeze-thawing that would have a good chance of reproductive success by establishing sperm mitochondria analysis as a standard parameter, in addition to existing standards. The idea is that mitochondrial parameters are not routinely added to the traditional CASA analysis when determining the odds that semen from a bull will lead to successful fertilization, embryo development and live offspring.  However, this idea is not new, since many articles have made the same proposition, using similar technologies and endpoints, including on bull sperm. For example, studies have performed the analysis of thawed bull sperm quality by assessing mitochondrial membrane potential (ΔΨm), mitochondrial activity (JC-1/PI), and other parameters such as acrosome integrity, DNA fragmentation, and antioxidant activity (SOD, CAT, MDA, GPx activity etc.). (e.g. Elkhawagah AR et al. Cryobiology 2020; Sun L et al. Cryobiology 2020; Kowalczyk A et al. Andrology 2021). Determining mitochondrial activity of frozen-thawed spermatozoa using Mitotracker (Deep Red 633) after freezing/thawing (PT) and after swim-up selection (SU), using flow cytometry has been reported two decades ago (e.g. by Hallap T et al. Theriogenology 63 (2005) 2311–2322).  Thus, if mitochondrial assessment is not performed routinely and only by limited numbers of breeders, it is more about financial considerations than lack of knowledge of the benefit of including mitochondrial parameters. 

Here, the authors divided the bull sperm samples in 2 groups, Group 1 = high AI and sperm quality (based on semen evaluation protocol for AI); vs group 2 = sperm samples disqualified from AI due to lower quality (such as low ejaculate volume, sperm concentration and motility, inflammation in reproductive tissues etc.).  All samples were analyzed by standardized Computer Assisted Sperm Analysis (CASA).  They further subdivided each group into an upper and a lower swim up fractions. They then assessed midpiece region mitochondria content with MitoTracker Green signal, and mitochondrial activity as reflected by the mitochondrial membrane potential (ΔΨm) with JC-1 method, mtDNA content, the transcript levels of Prdx1, Sod and Gss, and oxidative phosphorylation by measuring oxygen consumption rate.

The data showed that group 1 sperm contained more mitochondria than group 2 sperm, in relation to swimming ability. The difference between groups 1 and 2 for mitochondrial activity was more settled, but significant. The mtDNA copy number was also different between the groups and sub-groups. PRDX1 not significantly changed between 2 groups, but there was a good correlation in group 2 for SOD1 being higher in motile swimming sperm; and in group 1 for Gss, with upper swimmer having higher Gss levels than non-motile sperm. Finally, basal OCR and ATP-linked respiration were not different between groups. Thus, besides expected differences between groups 1 and 2, the authors had to examine the sub-categories based on swimming ability to find significant differences not clearly observed otherwise. This suggests that analyzing mitochondrial parameters is not sufficient in itself, and that separating swimmer and non-swimmer might actually be as good of a discriminatory parameter. Overall, the article is well-written with informative introduction and discussion sections, but most of the results presented were expected from published studies and the study is therefore not original.  Additional comments are below:

  1. The methods are well described but the numbers of samples is low (7 for group 1 and 10 for group 2). This means that each sub-set within group 1 or 2 had only few samples, but they did not provide this information. To assess the relevance of the data provided in the 4 sub-groups, one needs to know how many samples was in each of them. This information should be provided.
  2. The data presented between swimmers and non-swimmer spermatozoa in each group could be interesting to pursue. More specifically, finding possible defective genes or mechanisms in the failing sperm from group 1 could unveil new critical parameters explaining why some of the sperm in group 1 are failing despite good values in traditional tests. The finding that GSS expression between swimmer and non-swimmer in group 1 was different, but not the 2 other antioxidant genes PRDX1 and SOD1 suggests that a functional GSS is required for sperm movement, or it could be that loss of GSS is one of the reasons for loss of movement. Thus, the authors could design experiments questioning the role of GSS in sperm movement.

Similarly, for group 2 sperm judged as not adequate for AI, understanding why SOD1 was lower in the non-swimmer sperm, but not the other antioxidant genes, could help clarifying the role of SOD1 in sperm movement. Thus, designing experiments to establish the relationship between SOD1 activity and sperm movement could be interesting. For example, the authors could examine whether challenging sperm with H2O2 or other oxidative stressors while manipulating SOD1 levels or inhibiting it would alter sperm movement.

Author Response

(The authors gave the same response as above.)

Round 2

Reviewer 3 Report

The authors provided detailed responses to the questions and suggestions of the reviewer, putting more emphases on the importance of validating the adoption of in-depth analysis specifically applying to bull sperm quality, by including more mitochondrial parameters. Moreover, they stated that such validation will give incentive to AI stations for adopting better sperm assessment quality approaches. These comments are reasonable.

Only the last response appears to come from a misunderstanding of the suggestion the reviewer made to explore further how the decrease in GSS and SOD1 related to decreased sperm movement. Indeed, the reviewer did not suggest anywhere to examine ”transcript levels” of these antioxidant proteins, but rather the idea was to examine the integrity and functionality of these proteins. For example, the authors could quantify the protein levels of GSS and SOD1 in sperm from the different groups.

The other suggestion was to assess in vitro the ability of GSS or SOD1 to protect bull spermatozoa from H2O2-induced damages, by comparing the antioxidant responses of groups 1-4 control sperm and in sperm co-treating with inhibitors of these antioxidant enzymes together or sequentially with H2O2, in relation to sperm movement.  Again, this has nothing to do with RNA analysis! The idea here is that they could examine lipid peroxidation, ROS formation and/or protein modifications such as tyrosine nitration and S-glutathionylation in sperm from the different groups. There are well-established methods and existing kits available to assess these endpoints in sperm treated in vitro, and these approaches have been used for human and rodent spermatozoa (see for example “Morielli T and O'Flaherty C. Reproduction. 2015; 149:113-123”).

As the authors stated, validating that it is the case also in bull is important, even if it has been shown in other species. Moreover, this could help further characterizing differences between the sperm groups, and it could lead to the development of new assessment tools. Thus, it would have given more depth to the present study. In any case, it will be interesting to read more about it in the future.